# Evaluation of the Physicochemical and Sensory Characteristics of Different Fig Cultivars for the Fresh Fruit Market

**DOI:** 10.3390/foods9050619

**Published:** 2020-05-12

**Authors:** Cristina Pereira, Alberto Martín, Margarita López-Corrales, María de Guía Córdoba, Ana Isabel Galván, Manuel Joaquín Serradilla

**Affiliations:** 1Food Science and Nutrition, School of Agricultural Engineering, University of Extremadura, Avda. Adolfo Suárez s/n, 06071 Badajoz, Spain; crtpereiraj@gmail.com (C.P.); mdeguia@unex.es (M.d.G.C.); 2University Research Institute of Agro-Food Resources (INURA), Avda. de la Investigación s/n, Campus Universitario, 06006 Badajoz, Spain; 3Department of Horticulture, Research Centre Finca La Orden-Valdesequera (CICYTEX), Junta de Extremadura, Autovía Madrid-Lisboa s/n, 06187 Badajoz, Spain; margarita.lopez@juntaex.es (M.L.-C.); anabelgalvanromero@gmail.com (A.I.G.); 4Department of Plant Sciences, Agrifood Technology Institute of Extremadura (INTAEX-CICYTEX), Junta de Extremadura, Avda. Adolfo Suárez s/n, 06007 Badajoz, Spain; manuel.serradilla@juntaex.es

**Keywords:** *Ficus carica* L., sensory properties, volatile compounds, colour, firmness

## Abstract

Physicochemical and sensory properties of nine fig cultivars: ‘San Antonio’ (SA), ‘Blanca Bétera’ (BB), ‘Brown Turkey’ (BT), ‘Tres Voltas L’Any’ (TV), ‘Banane’ (BN), ‘Cuello Dama Blanco’ (CDB), ‘Cuello Dama Negro’ (CDN), ‘Colar Elche’ (CE), and ‘De Rey’ (DR), were compared at three different ripening stages. Weight, size, titratable acidity, pH, skin and flesh colours, firmness, maturation index (MI), and volatile compounds were determined in samples from two consecutive seasons, in addition to both descriptive and hedonic sensory analysis. The mechanical behaviour of figs determined by firmness analysis and colour changes in both skin and flesh was the most important trait for the discrimination of ripening stages. Notable differences among cultivars were found in most of the parameters studied, in particular the inter-cultivar differences highlighted for MI, pH, acidity, and skin colour analyses, followed by volatile compounds. Principal component analysis (PCA) indicated that MI, pH, colour parameters of flesh (h and L*), and terpenes were the best physicochemical indices to determine overall acceptability which is highly correlated with the sensory attributes flesh colour and fruit flavour. The results suggested that CDN and SA showed huge consumer acceptability among the fig cultivars studied.

## 1. Introduction

The fig (*Ficus carica* L.) constitutes a spice that is widely grown in the Mediterranean area, where the fig tree population has been present since its domestication [1]. In this area, both fresh and dried figs are an important part of the diet, being especially rich in nutrients such as sugar, fibre, proteins, and minerals, but also in organic acids and polyphenols. Spain is the major producer of figs in Europe, with approximately 36,380 tonnes, i.e., 38% of European production and 3% of world production [2]. Although most commercial production is of dried fruit, figs are also widely consumed as fresh fruit.

Fresh fruit quality is determined by nutritional and bioactive composition, but also by other parameters related to the sensory characteristics, including firmness, visual appearance, taste, and aroma [3]. Firmness is one of the primary attributes determining consumer acceptance, flesh firmness being the parameter used to determine the harvest time as well as the maturity grade during postharvest of perishable fruit such as fig [4]. Sugars, acids, and phenolic compounds contribute to the taste and colour of figs, but also to the characteristic of flavour, which is dependent mainly on the proper balance of the volatile chemical constituents [5,6]. On the other hand, the aromatic compound profile of each variety is considered to be unique and has a great influence on organoleptic characteristics and therefore on consumer acceptance [3,5]. Recently, the use of techniques as solid-phase microextraction of headspace (HS-SPME) with gas chromatography analysis with a mass detector (GC-MS) has made it possible to identify and quantify individually this complex mixture of aromatic compounds, which mainly includes compounds such as alcohols, aldehydes, ketones, esters and terpenoids. Additionally, this complex mixture depends on several factors such as soil, climate, genotype, ripeness, and technological aspects [7]. Compounds such as ethyl acetate, hexanal, β-caryophyllene, limonene, (E)-2-hexenal, and octanal have been attributed primarily to the aroma of fresh figs [8,9,10]. In addition, compounds such as furfural, benzaldehyde, phenol, among others, have also shown a remarkable influence on the aroma of fresh fig [11,12]. The volatile composition in fresh figs varies during the ripening process and with it the perception of their sensory characteristics [5,11]. Thus, in fruit such as kiwifruit, a considerable increase in the ester content and a decrease in the concentration of aldehydes throughout ripening has been described [13]. However, in the case of figs, the concentration of both compounds has also been clearly influenced by genotype [1].

The majority of studies related to the characterisation of quality parameters of fresh fig cultivars approach specific quality aspects such as physicochemical properties [14,15,16], firmness, or aroma compounds [5,10,11]. Regarding the volatile aroma profile of figs, some studies have focused on the analysis of aromatic compounds from leaves, spirits, extracts and others [5,8,9,10,11]. So far, however, only a few studies showed an overall view of the relation among physicochemical parameters and sensory attributes taking into account different ripening stages. Thus, Crisosto et al. [17] reported the influence of two ripening stages on different physicochemical traits as well as other parameters such as the total antioxidant capacity of four fig cultivars currently grown in California, highlighting the great impact of total soluble solids content on consumer acceptance. King et al. [18] also characterised the sensory properties of 12 California-grown fresh fig cultivars from six different sources, finding significant correlations between sensory and physicochemical data. These authors highlighted the importance of selecting cultivars with strong flavours that remain firm as they mature.

To our knowledge, a comprehensive and interannual study of both physicochemical and sensory quality characteristics in several commercial cultivars of fresh fig has never been carried out. In this research, we established the relation among these physicochemical and sensory parameters and an overview of quality traits of the fig cultivars studied for fresh fruit market, taking into account the changes associated with the different commercial ripening stages.

## 2. Materials and Methods

### 2.1. Plant Material and Experimental Design

The nine fig tree cultivars studied for fresh consumption in the order of ripening (early, middle or late) were ‘San Antonio’(SA), ‘Blanca Bétera’ (BB), ‘Brown Turkey’(BT), ‘Tres Voltas L’Any’(TV) as early cultivars, ‘Banane’(BN) as mid cultivar, ‘Cuello Dama Blanco’ (CDB), ‘Cuello Dama Negro’(CDN), ‘Colar Elche’(CE), and ‘De Rey’ (DR) as late cultivars. Studies of morphological and molecular characterisation conducted in Spanish germplasm fig indicate that ‘Cuello Dama Blanco’ and ‘Colar Elche’ are the same varieties as ‘Kadota’ and ‘Mission’, respectively [19,20,21]. All cultivars were selected among those available in the national germplasm bank of the fig tree is located in the research centre “Finca La Orden-CICYTEX” at an altitude of 223 m above sea level (latitude 38°85′19″ N, longitude −6°68′28″ W, Guadajira, Badajoz, Spain) based on parameters of fruit quality. Figs were harvested manually from July to October. With respect to the experimental design, fig samples were hand-collected when they were fully mature and were harvested during two consecutive seasons from an experimental orchard established in 2007 and previously described in our previous work [22].

For the maturation study, three different ripening stages of each cultivar were selected according to their skin colour and firmness as per expert harvester criteria. Stage 1 corresponded to the greener fruit, whereas Stage 3 corresponded to mature fruit. For each physicochemical determination, three replicates of 10 fruit for each ripening stage and cultivar were established per year. All analyses were conducted using fresh fruit, but for analysis of volatile aroma profile, the samples were weighed in vials and stored at −80 °C for later analysis.

### 2.2. Weight and Size

Both parameters were determined using an AE-166 balance (Mettler, Madrid, Spain) for weight (g) and a DL-10 digital micrometre (Mitutoyo, Kawasaki, Japan) for size (mm).

### 2.3. Total Soluble Solids (TSS), Titratable Acidity (TA), pH, and Maturation Index (MI)

A model RM40 Mettler Toledo digital refractometer at 20 °C was used to measure total soluble solids (TSS) in °Brix. On the other hand, 5 g aliquots of fig homogenate diluted to 50 mL with deionised water from a Milli-Q water purification system (Millipore, Bedford, MA, USA) were used to determine titratable acidity and pH, using a T50 Compact Stirrer for automatic titration (Mettler Toledo, Madrid, Spain), titrating up to pH 7.8 with 0.1 mol L^−1^ NaOH and expressing the results in g citric acid 100 g^−1^ fresh weight (FW).

The maturation index (MI) was calculated as described by Pereira et al. [22].

### 2.4. Colour

Skin and flesh colour of figs was measured according to Pereira et al. [22]. The parameters brightness (L*), chroma (C*) and hue angle (h*) were measured using a Konica Minolta CM600 spectrophotometer in accordance with the CIELab system.

### 2.5. Firmness

A 6% deformation was applied by a 70 mm aluminium plate coupled to a TA.XT2i Texture Analyser (Stable Micro Systems, Godalming, UK) to measure firmness in N mm^−1^ [22].

### 2.6. Determination of Volatile Compounds

The volatile profile from each ripening stage and cultivar was analysed by solid-phase microextraction (SPME) with a 10 mm-long, 75 µm-thick fibre coated with Carboxen™/polydimethylsiloxane (Supelco, Bellefonte, PA) as described by Serradilla et al. [23].

The volatile compounds were identified and semi-quantified using an Agilent 6890 GC/5973 MS system (Agilent Technologies) using a DB-5 (Agilent Technologies J&W, Santa Clara, CA, USA) bonded fused silica capillary column, coated with 5% phenyl/95% polydimethylsiloxane (30 m × 0.32 mm inner diameter, 1.05 μm film thickness). For the identification of volatile profile, in addition to using the NIST/EPA/NIH mass spectrum library (comparison quality > 90%), and Kovats indices, which were calculated using a mixture of n-alkanes (R-8769, Sigma Chemical Co., St. Louis, MO, USA) run under the same conditions [24], pure compounds under the same chromatographic conditions were also used to confirm the identifications.

### 2.7. Sensory Analysis

For the sensory analysis, only samples from ripening Stages 2 and 3 were used. A trained panel of 10 panellists, 6 women and 4 men between the ages of 30 and 50 years old, was used to carry out a descriptive sensory analysis, assessing the parameters previously described in our previous work [12]. A numbered scale from 1 to 10 points was used. Each panellist evaluated a total of two defect-free fruit per ripening stage and cultivar immediately after harvest and after reaching a flesh temperature of 20 °C under white lighting, airflow, and temperature (20–22 °C) controlled conditions. Samples were presented to each panellist on plastic plates in random order using a 3-digit code for each sample, assessing the following sensory attributes: external appearance, skin colour, flesh colour, firmness, sweetness, acid, bitter, juiciness, presence of seeds, and fruit flavour. One session was conducted per week throughout the harvest period. Additionally, hedonic tests, using a numbered scale from 1 to 10 points, were performed to evaluate overall acceptability with a total of 65 untrained consumers, but regular fig consumers, consisting of 35 women and 20 men aged 20 to 50 years old, was tested every 15 days throughout the harvest period.

### 2.8. Statistical Analysis

SPSS for Windows, 19.0 (SPSS Inc. Chicago, IL, USA) was used to carry out an analysis of variance (ANOVA) of the mean values of physicochemical parameters, the area of volatile compounds, and sensory characteristics. Tukey’s honestly significant differences (HSD) test (*p* ≤ 0.05) was applied to separate means. Principal component analysis (PCA) was used to analyse the relationships among the parameters studied, using ‘ripening stage’ and ‘cultivar’ as classification variables.

## 3. Results and Discussion

### 3.1. Physicochemical Properties

Fig cultivar weights and sizes at the three ripening stages are shown in Figure 1A,B. The mean weight of the cultivars studied was 46.5 g, with BT and BN showing the highest values for this parameter. These same cultivars, along with CE and BB, showed a significant weight increase with the ripening process, mainly between Stage 1 and Stage 2. For BT, mean weight increased from 54.8 g (Stage 1) to 77.5 g (Stage 3). A similar weight increase associated with maturity stage was also found for this cultivar by Crisosto et al. [17]. On the contrary, TV presented weights less than 30 g for the three stages studied. The weight tendency in fig cultivars was also observed for size, with 43.2 mm as the mean value, although in this case, differences among maturity stages were not found (Figure 1B).

The pH values of the fig cultivars studied are given in Figure 1C. The mean pH value was 5.8, ranging from 5.16 (BB; Stage 1) to 6.39 (SA; Stage 3), which are slightly higher than those described by other authors [25,26]. There was an increase in the pH values with increasing maturity, which resulted in a less acid product. This change associated with ripening time was more evident for cultivars BN, TV, and BB with a decrease of TA of more than 0.6 g citric acid 100 g^−1^ FW from Stage 1 to Stage 3 (Figure 1D). The mean value of this parameter for the cultivars studied was 1.2 g citric acid 100 g^−1^ FW, ranging from 0.72 (Stage 3 of SA) to 2.14 g citric acid 100 g^−1^ FW (Stage 1 of CDN). This TA ratio is in line with those obtained by Çalişkan and Polat [27] for fig genotypes grown in the Eastern Mediterranean Region of Turkey as well as some Turkish cultivars. In general, the early SA cultivar was characterised by exhibiting the highest pH values and the lowest TA values at the most advanced stages of ripening.

The fig cultivars studied had an average °Brix of 20.4 and MI of 200.6, all of them showing an evident increase for these parameters during the ripening process of fig fruit, with the highest values at maturation Stage 3 (Figure 1E,F). The TSS/acid ratio is directly related to fruit taste and in the fig’s aptitude for the drying process [27]. The differences between Stage 1 and Stage 3 were more than 6 °Brix for late cultivars CDN and CE, whereas the variation for mid cultivars BN and CDB was less than 3 °Brix. For MI, the more relevant differences between ripening stages were found for early and mid cultivars SA, BB, and BN, in contrast to late cultivars DR and CDB, which showed no significant differences. In general, the °Brix values found in our study were in agreement with other reports [26,28,29], whereas the MI values were slightly higher, mainly due to the low acidity presented in the studied cultivars.

### 3.2. Colour and Firmness

The colour parameters of the fig varieties studied are shown in Figure 2. There were evident differences in the mean values of skin colour parameters between the dark (CDN and CE), purple/yellow (SA, BT, and DR), and green cultivars (CDB, TV, BN, and BB), but also between ripening stages. An increase of h* values was observed for CDN and CE at Stage 3, whereas for the rest of the varieties the decrease of L* values was the more relevant change. Crisosto et al. [17] described a significant increase of h* values at the higher maturity stage studied for cv. Mission (syn. CDN and CE). Likewise, differences were also observed in flesh colour parameters among different cultivars of figs (*p* > 0.05), mainly related to the coordinates L* and h* (Figure 2B). The cultivars CDB and SA showed the highest values for both colour parameters mentioned, with L* values of 55.53 and 59.19 and h* values of 73.47 and 71.39, respectively. On the contrary, the lowest values of L* and h* in fig flesh were found for CDN and BT (lower than 46.39 and 48.06 for L* and 46.28 and 46.91 for h*, respectively). These results agree with those for some Turkish fig cultivars which showed great variability among cultivars with mean values of 53.79 for L* and 42.37 for h* [14].

On the other hand, there were also differences in the flesh colour among the three ripening stages of the fig cultivars studied. In this case, C* values decreased during the ripening process for all fig cultivars studied as a consequence of flesh darkening due to the accumulation of anthocyanins, this change being more evident in the dark-skinned cultivars CDN and BT. A negative correlation between total anthocyanins in flesh fruit and the chromatic parameter chroma has been described for several fruit such as sweet cherry [30]. After the visual appearance, firmness is the most relevant factor that determines the acceptability of fleshy fruit such as figs [4], firmness being a relevant component. Firmness values decreased during the maturation of figs, this process being cultivar-dependent (Figure 3). Initially, the mean fig firmness values were 2.57 N mm^−1^ at Stage 1, decreasing to 0.75 N mm^−1^ at Stage 3. Our results during the ripening process of fig cultivars were similar to those obtained by other authors [17,22]. Regarding cultivar differences, the firmness values of SA and BB were higher than those of most of the cultivars studied, whereas TV presented firmness values significantly lower than the rest of the cultivars for all ripening stages studied.

### 3.3. Volatile Compounds

A total of 68 compounds were identified in the fig cultivars studied using HS/SPME and GC/MS (Table 1). These volatile compounds were classified as aldehydes (20), hydrocarbons (9), furans (8), alcohols (4), terpene compounds (4), ketones (3), acids (3), esters (3), pyranone derivates (2), pyrimidines (1), and ethers (1). Similar findings in fresh figs were described in previous studies [5,10,12,31,32].

Among the main volatile compounds that define fresh figs’ aroma profile are aldehydes [31]. These compounds represent a mean percentage of 18.25% of the total area, including linear, branched, and aromatic aldehydes (Table 1). Regarding aromatic aldehydes, benzaldehyde (AL12) accounted for 7.13% of the total area as mean value for the fig cultivars studied, being the most abundant aldehyde found. They originate from the shikimic acid pathway and contribute greatly to the characteristic aroma of fresh figs [33]. The main linear aldehydes identified were (E)-2-hexenal (AL9), nonanal (AL18), and hexanal (AL8) (2.65%, 2.41%, and 1.76% of the total area, respectively), which have also been reported to be key to the volatile aroma profile of some fig cultivars [12,31]. These compounds exceeded 5% of the total area for some cultivars studied and are characterised by exhibiting green leaf notes [34]. Finally, among the branched aldehydes, 2-methyl-2-butenal (AL5) showed a high percentage in most of the cultivars studied (1.09% of the total area), while the relative concentration of the other branched aldehydes was less than 1% of the total area.

Hexane (H3) was the most abundant hydrocarbon, a chemical class that involved 3.18% of the total area as mean value for these studied cultivars, including branched and aromatic compounds (Table 1). In addition, short-chain alkanes have also been reported, although in lower concentration, to be present in fig fruit [10], described as non-contributors to fruit flavour. On the contrary, particularly in light and yellow-green cultivars, other relevant compounds in the aroma profile of figs are alcohols (2.50% of the total area), whether linear, branched, or aromatic. (Table 1) [31]. 3-Heptanol (OL2) is associated with fresh green odours and green leaf notes that are typically linked to fruit such as yellow passionfruit [35].

Furans represented 23.05% as a mean percentage, ranging significantly between 1.83% and 57.33% of the total area, mainly due to the variability found for 5-hydroxymethylfurfural in the cultivars studied (Table 1). Both 5-Hydroxymethylfurfural and furfural have been associated with sweet flavour notes and indeed both have been identified among the characteristic aroma compounds of several other fruit, such as kiwifruit [6]. Additionally, furans and the derivates of pyranones have been reported to be derived directly from carbohydrates [36]. Pyranones were the fourth most abundant chemical class in the cultivars studied, representing a mean percentage of 12.65% of the total area. 3-Hydroxy-2,3-dihydromaltol (P1) was characterised by being the dominant compound within this family with 11.06% of the total area for these studied cultivars, describing its aromatic note as caramel. Furan and pyranone derivates show an outstandingly low odour threshold and are also considered to be primary contributors to the volatile profile of dried figs [11,37].

Regarding ketones (3.24% of the total area), a total of three of these compounds were identified (Table 1). 3-Heptanone was the ketone detected at highest concentrations in the cultivars studied. Pino et al. [38] considered this compound as a green fatty aroma note, and it has been described among the most important aroma-active volatiles of fruit such as choch (*Lucuma hypoglauca* Standley).

With respect to acids (3.12% of the total area), acetic acid (AC1), nonanoic acid (AC3), and the most abundant 2-ethyl hexanoic acid (AC2) were detected (Table 1). This last compound has a negative effect on the overall aroma of fruit derivates, possessing an unpleasant odour with slightly putrid notes [39]. 2-Ethylhexanoic acid has been also reported as a regular food packaging material contaminant [40]. In our study, cultivars showed a mean percentage of 2.66% of the total area for this acid.

Esters, with 26.93% of the total area, represented the other main group of aromatic compounds detected in these fruit. These compounds are generated from the esterification of alcohols and acyl-CoA derivates, highlighting the concentration shown by ethyl acetate (ES2), at 26.53% of the total area, as the mean value in the cultivars studied (Table 1). This result could suggest that this volatile compound may be relevant to the aroma profile of these cultivars. This compound was also described in two Portuguese fig varieties (‘Branca Tradicional’ and ‘Pingo do Mel’), although the concentration shown by these cultivars was lower than that obtained in this study [31].

Other compounds, which represented around 1% of the total area of volatile compounds of the cultivars studied, were monoterpenes such as α-pinene (T1), limonene (T3), and linalool (T4) (Table 1), which are among the most frequent groups of aroma compounds identified in figs [5,8,11,33].

In general, remarkable fluctuations were observed in the volatile profiles of the samples studied according to the values of standard deviations and ranges shown in Table 1. To understand the role of both factors, cultivar and ripening stage, in this variability, a PCA was performed with the major volatile compounds (>0.85% of total area). The PCA showed clear differences in the volatile profile among the cultivars, and a limited influence of the ripening stages selected in this work (Figure 4 and Figure 5). Thus, the cultivars SA, BT, and BN showed a lower amount of the main volatile compounds compared to DR, BB, and CDB. Concretely, the cultivar DR was associated with high concentrations of the main furans (F1, F2, F3, F7, and F8), pyranones (P1 and P2), and, to a lesser extent, aldehydes such as benzaldehyde (AL13) and hexanal (AL8). In the case of cultivars BB and CDB, their volatile profiles are highlighted for the high amount of 2-methyl-2-butenal (AL5), decanal (AL20), linalool (T4), and hexane (H3) among other compounds (Figure 4 and Figure 5). The differences found in both physicochemical properties and volatile profiles of the fig cultivars studied may have a relevant impact on their sensory parameters.

### 3.4. Sensory Analysis

ANOVA of sensory descriptors (Table 2) shows significant differences among most of the fig cultivars studied but not between ripening Stages 2 and 3, with the exception of external appearance which was better for samples of ripening Stage 2. CDN and CE cultivars obtained the best scores (higher than 7.00) for the descriptive parameters—external appearance, skin colour, and texture—whereas BB was scored highest by panellists for sweetness (5.56), although no significant differences were found for this parameter. The highest scores for fresh colour and fruit flavour descriptors were found in CDB and SA samples (5.97–6.70 and 5.67–6.25, respectively), both parameters showing a high degree of correlation with acceptability in the hedonic test. In fact, these cultivars showed the best scores for the overall acceptability descriptor (6.71 and 6.65). In the case of SA, the high score for the descriptor fruit flavour was not correlated with the relatively low number of volatile compounds found for this cultivar (Figure 4 and Figure 5), showing that the flavour is a complex combination of not only olfactory but also gustatory-tactile and kinaesthetic sensations [41]. On the contrary, BT presented the worst score for acceptability (4.37), which was clearly related to its lowest scores for skin colour (5.04), fresh colour (4.02), and fruit flavour (4.05). The difference between these results and those found by other authors for BT can be partially explained by the better sensory quality of pollinated fruit with respect to the parthenocarpic fruit used for our study [42].

### 3.5. Interactions between the Analytical Parameters and Sensory Characteristics

PCA was carried out for the whole set of data to obtain an interpretable overview of the main information. Samples at ripening Stage 1 were excluded as they were not sensorially analysed. Figure 5 shows the two-way loadings and score plots, where PC2 and PC3 were plotted against PC1 to show a high percentage of the total variance (48.20–41.04%). High values for MI, pH, skin C*, and skin L* were explained by the positive axis of PC1 and were related to SA and, to a lesser extent, to CDB and BB. These parameters showed a negative correlation with acidity, skin h, and the sensory descriptors acid taste, skin colour, external appearance, and texture, which were highlighted in CDN and CE. The second PC was mainly explained by most of the chemical families of the volatile compounds located in the extreme of the positive axis, relating high values of those to the cultivar DR. On the contrary, the cultivars with the highest weight and calibre values, BT and BN, were associated with the negative axis of PC2 and therefore with a poor concentration of volatile compounds. Variability of the sensory descriptors sweetness and juiciness was mainly explained by the positive axis of PC3, showing a clear negative correlation with the descriptor bitter taste, high values being those associated with BB.

Regarding acceptability, high scores in the hedonic test were correlated with MI, pH, flesh h, and terpenes, in addition to the above-mentioned sensory descriptors (flesh colour and fruit flavour). Likewise, the association of high acceptability scores with CDB (plots defined by PC1 and PC2) was again observed, but also with SA (plots defined by PC1 and PC3).

## 4. Conclusions

In conclusion, the characteristics of the nine fig cultivars included in this study can be explained on the basis of the physicochemical and sensory properties studied, most of them showing notable differences among them. This confirms the relevance of characterising fig cultivars for the assessment of potential consumer acceptability. In agreement with the PCA results, MI, pH, acidity, and skin colour show the highest variability among the parameters studied in fresh figs, followed by volatile compounds. Acceptance of the cultivars studied is associated with high values for MI, pH, flesh colour parameters (h and L*), and terpenes, highlighting the sensory attributes flesh colour and fruit flavour. In particular, CDN and SA showed high consumer acceptability among the fig cultivars studied.

## Figures and Tables

**Figure 1 foods-09-00619-f001:**
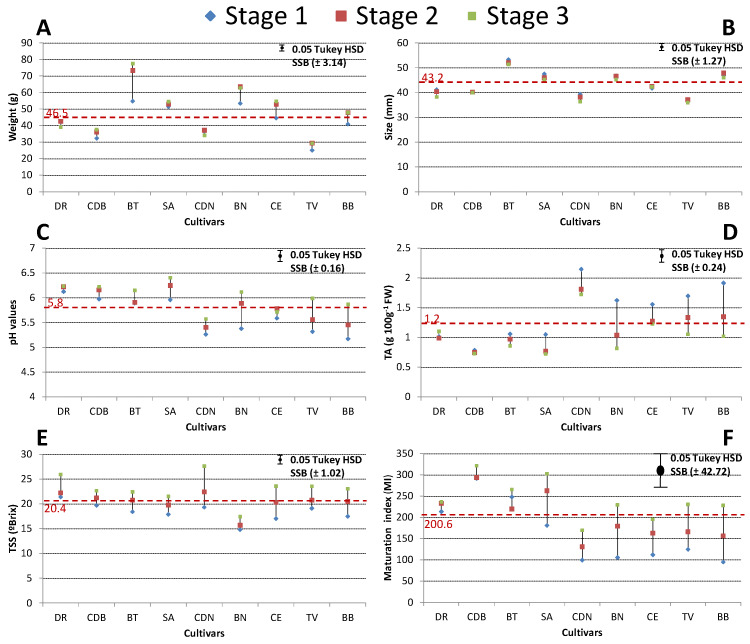
Physicochemical parameters of the fig cultivars at the three ripening stages studied. Weight (**A**); size (**B**); pH (**C**); titratable acidity (TA) (**D**); total soluble solids (TSS) (**E**); maturation index (MI) (TSS/TA) (**F**). SA, San Antonio; BB, Blanca Bétera; BT, Brown Turkey; TV, Tres Voltas L’Any; BN, Banane; CDB, Cuello Dama Blanco; CDN, Cuello Dama Negro; CE, Colar Elche; DR, De Rey. Tukey HSD, Tukey’s honestly significant differences; SSB, Statistical significance bar.

**Figure 2 foods-09-00619-f002:**
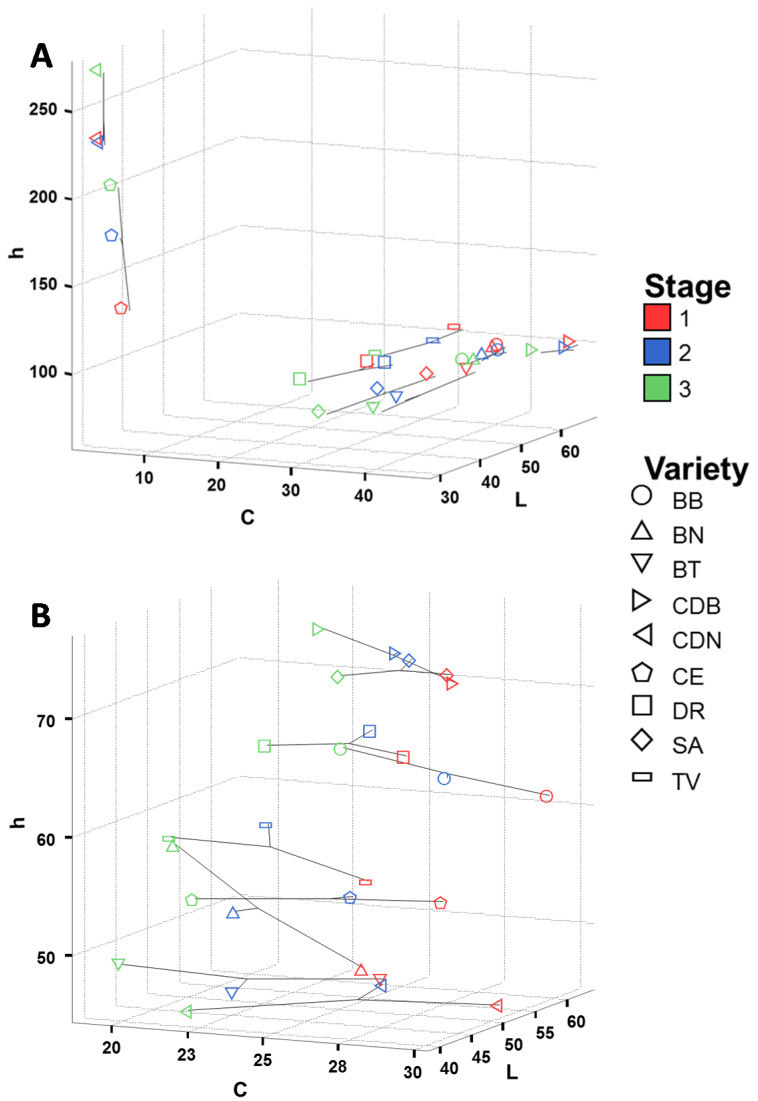
Colour parameters of the fig cultivars at the three ripening stages studied. Skin (**A**); flesh (**B**). L*, Brightness; C*, Chroma and h*, Hue angle.

**Figure 3 foods-09-00619-f003:**
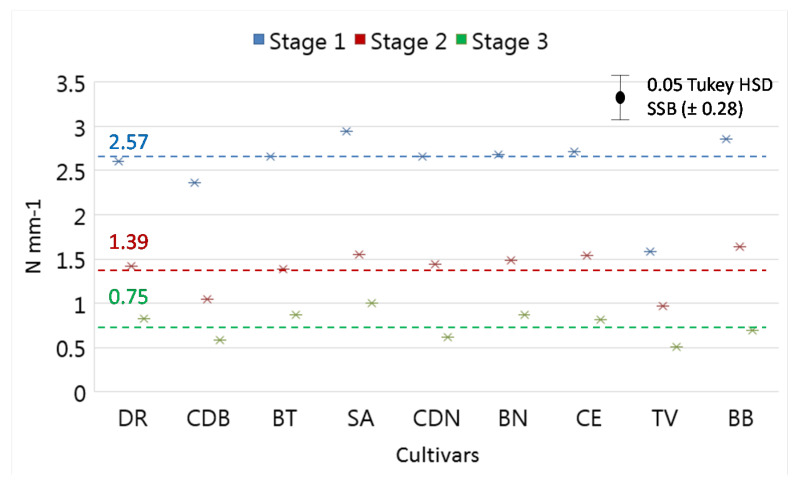
Firmness values of the fig cultivars at the three ripening stages studied.

**Figure 4 foods-09-00619-f004:**
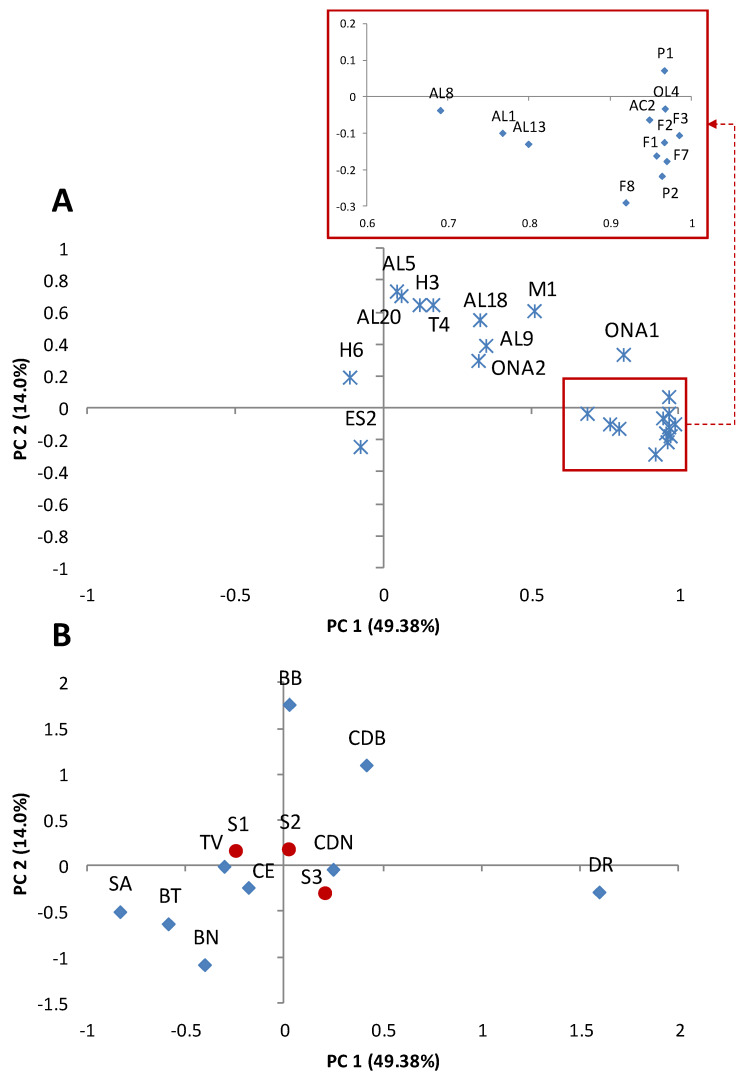
Loading plot (**A**) and score plot (**B**) after principal component analysis of the varieties, ripening stages, and volatile compounds in the plane by two first principal components (PC1 and PC2). Code letters for volatile compounds are shown in Table 1.

**Figure 5 foods-09-00619-f005:**
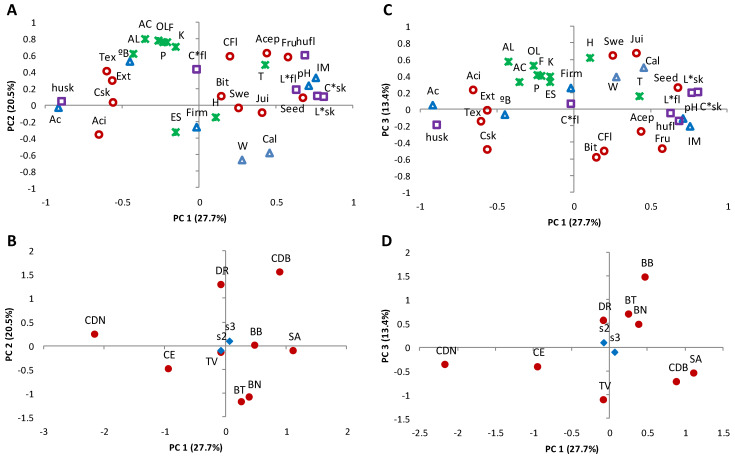
Loading plots (**A**,**C**) and score plots (**B**,**D**) after principal component analysis of the varieties, ripening stages, physicochemical, and sensory parameters and in the planes by three first principal components (PC1, PC2, and PC3). Physicochemical parameters (△), weight (W), calibre (C), °Brix (°B), acidity (Ac), pH, maturation index (IM), firmness (Firm). Colour parameters (□), skin h* (hsk), C* (C*sk), L* (L*sk); fesh h* (hfl), C* (C*fl), L* (L*fl). Chemical families of volatile compounds (ᚕ), aldehydes (AL), acids (Ac), alcohols (OL), ketones (K), furans (F), pyrazines (P), hydrocarbons (H), terpenes (T), esters (ES). Sensory parameters (○), external appearance (Ext), skin colour (Csk), fresh colour (CFl), texture (Tex), sweetness (Swe), bitterness (Bit), juiciness (Jui), fruit flavour (Fru), overall acceptability (Acep).

**Table 1 foods-09-00619-t001:** Volatile compounds in the fresh fig samples studied.

Volatile Compounds	CD ^†^	KI ^‡^	ID ^§^	RT ^¶^	AAU ^|^	Percentage of Area (%) ^#^			
						Mean	SD	Min	Max
*Hydrocarbons*	*H*				*452*	*3.18*	*3.20*	*0.32*	*15.15*
Pentane, 2-methyl-	H1	570	A	7.0	2	0.02	0.10	0.00	0.52
Pentane, 3-methyl-	H2	585	A	7.7	6	0.05	0.20	0.00	1.01
**Hexane ^¤^**	**H3**	**600**	**A**	**8.9**	**241**	**1.84**	**1.64**	**0.23**	**7.29**
Cyclopentane, methyl-	H4	635	B	10.3	4	0.03	0.13	0.00	0.67
Heptane	H5	700	A	14.7	10	0.08	0.18	0.00	0.55
**Toluene**	**H6**	**779**	**B**	**18.8**	**106**	**0.66**	**1.32**	**0.00**	**5.57**
Ethylbenzene	H7	864	B	23.6	17	0.10	0.21	0.00	0.86
p-Xylene	H8	869	B	23.8	52	0.29	0.64	0.00	2.39
Tetradecane	H9	1400	A	39.4	14	0.10	0.27	0.00	1.16
*Alcohols*	*OL*				*378*	*2.50*	*1.18*	*0.63*	*5.71*
3-Buten-1-ol, 3-methyl-	OL1	726	B	16.7	15	0.11	0.13	0.00	0.53
3-Heptanol	OL2	894	B	24.7	44	0.40	0.71	0.00	2.66
Branched alcohol	OL3	1028	D	29.8	26	0.30	0.49	0.00	1.90
**Aromatic alcohol**	**OL4**	**1080**	**D**	**31.7**	**292**	**1.69**	**0.81**	**0.40**	**3.40**
*Aldehydes*	*AL*				*2407*	*18.25*	*10.34*	*4.99*	*58.98*
**Propanal**	**AL1**		**C**	**5.2**	**90**	**0.60**	**0.42**	**0.00**	**1.95**
2-Butenal, (E)-	AL2	640	B	11.6	5	0.03	0.09	0.00	0.39
Butanal, 3-methyl-	AL3	645	A	11.9	6	0.04	0.10	0.00	0.39
Butanal, 2-methyl-	AL4	660	A	12.5	83	0.42	0.64	0.00	2.51
**2-Butenal, 2-methyl-**	**AL5**	**745**	**B**	**17.5**	**98**	**0.94**	**1.09**	**0.00**	**3.59**
2-Pentanal, (E)-	AL6	750	B	18.0	27	0.24	0.45	0.00	1.89
2-Butenal, 3-methyl-	AL7	783	B	19.4	53	0.37	0.22	0.00	0.72
**Hexanal**	**AL8**	**800**	**A**	**20.2**	**221**	**1.76**	**1.44**	**0.22**	**5.98**
**2-Hexenal, (E)-**	**AL9**	**853**	**A**	**22.8**	**305**	**2.65**	**2.97**	**0.50**	**15.15**
Heptanal	AL10	902	A	25.1	58	0.28	0.40	0.00	1.72
2,4-Hexadienal (E,E)-	AL11	910	B	25.4	1	0.02	0.07	0.00	0.37
2-Heptenal	AL12	952	B	27.4	4	0.04	0.06	0.00	0.23
**Benzaldehyde**	**AL13**	**956**	**A**	**27.8**	**985**	**7.13**	**6.33**	**1.45**	**32.44**
Octanal	AL14	1004	A	29.1	50	0.35	0.20	0.00	0.76
2,4-Heptadienal	AL15	1010	B	29.4	14	0.11	0.24	0.00	1.09
Benzeneacetaldehyde	AL16	1051	B	30.7	20	0.14	0.38	0.00	1.89
2-Octenal, (E)-	AL17	1062	C	31.1	16	0.14	0.22	0.00	0.91
**Nonanal**	**AL18**	**1106**	**A**	**32.6**	**294**	**2.41**	**1.74**	**0.00**	**8.22**
2-Nonenal	AL19	1164	B	34.5	7	0.04	0.07	0.00	0.27
**Decanal**	**AL20**	**1204**	**A**	**35.9**	**70**	**0.54**	**0.60**	**0.00**	**2.67**
*Ketones*	*K*				*495*	*3.24*	*1.70*	*0.71*	*6.09*
**3-Heptanone**	**K1**	**889**	**B**	**24.3**	**373**	**2.52**	**1.41**	**0.67**	**6.09**
**1,3-Cyclopentanone**	**K2**	**992**	**C**	**28.8**	**106**	**0.64**	**0.57**	**0**	**1.89**
2-Cyclopenten-1-one, 2-hydroxy-3-methyl- (Corylon)	K3	1029	C	30.0	15	0.08	0.10	0	0.34
*Acid*	*AC*				*594*	*3.12*	*1.77*	*0.02*	*7.28*
Acetic acid	AC1			8.3	10	0.05	0.12	0	0.51
**Hexanoic acid, 2-ethyl-**	**AC2**	**1123**		**32.8**	**520**	**2.66**	**1.62**	**0**	**6.15**
Nonanoic acid	AC3	1277		37.3	64	0.41	0.35	0	1.13
*Ester*	*ES*				*3545*	*26.93*	*20.31*	*0.69*	*65.61*
Acetic acid, methyl ester	ES1	554	A	6.2	42	0.38	0.30	0.01	1.15
**Acetic acid, ethyl ester**	**ES2**	**628**	**A**	**9.6**	**3496**	**26.53**	**20.11**	**0.67**	**65.26**
Butanoic acid, methyl ester	ES3	723	B	16.4	7	0.02	0.10	0	0.52
*Furans*	*F*				*5831*	*23.05*	*13.72*	*1.83*	*57.33*
**Furfural**	**F1**	**830**	**A**	**21.9**	**966**	**4.24**	**2.32**	**0.51**	**8.21**
**2-Furanmethanol**	**F2**	**859**	**B**	**23.0**	**903**	**4.39**	**2.06**	**0.97**	**8.2**
**2(3H)-Furanone, 5-methyl-**	**F3**	**930**	**B**	**26.0**	**336**	**1.69**	**0.83**	**0.1**	**2.88**
Furaneol	F4	1058	B	30.8	49	0.30	0.60	0	2.31
Unknown furan 1	F5	1098	D	32.2	156	0.45	0.64	0	2.31
5-(Hydroxymethyl)-2(5H)-furanone	F6	1178	B	34.9	65	0.33	0.57	0	2.34
**Unknown furan 2**	**F7**	**1197**	**D**	**35.5**	**509**	**2.14**	**1.34**	**0**	**4.8**
**5-Hydroxymethylfurfural**	**F8**	**1224**	**B**	**36.5**	**2847**	**9.52**	**8.41**	**0**	**33.38**
*Pyranones*	*P*				*3041*	*12.65*	*7.15*	*0.81*	*26.64*
**2H-Pyran, 3,4-dihydro-**	**P1**			**25.5**	**313**	**1.59**	**0.76**	**0.4**	**2.97**
**3-Hydroxy-2,3-dihydromaltol**	**P2**	**1140**	**B**	**34.2**	**2727**	**11.06**	**6.54**	**0.36**	**25.06**
*Monoterpenes*	*T*				*156*	*1.17*	*1.46*	*0.00*	*4.72*
α-Pinene	T1	940	B	26.7	9	0.05	0.14	0	0.65
Unknown monoterpene	T2	1006	D	29.2	9	0.04	0.11	0	0.39
Limonene	T3	1030	A	30.3	38	0.19	0.29	0	1.18
**Linalool**	**T4**	**1098**	**B**	**32.6**	**101**	**0.89**	**1.41**	**0**	**4.35**
*Miscellaneous*									
Ethyl ether	ET1		A	5.3	10	0.07	0.19	0	0.91
**Thymine**	**M1**	**1075**	**B**	**31.6**	**855**	**5.85**	**3.28**	**0.58**	**11.5**

^†^ CD: compound code used. ^‡^ KI: Kovats retention index. ^§^ ID: reliability of identification: A, identified by a comparison to standard compounds; B, tentatively identified by the NIST/EPA/NIH mass spectrum library (comparison quality >90%) and Kovats retention index; C, tentatively identified by the NIST/EPA/NIH mass spectrum library (comparison quality >90%); D, tentatively identified by the NIST/EPA/NIH mass spectrum library (comparison quality <90%). ^¶^ RT: retention time. ^|^AAU: arbitrary area units. ^#^ %: relative abundance. ^¤^ In bold: volatile compound included in PCA analysis, according to high relevance in the volatile profile of the fig cultivar studied.

**Table 2 foods-09-00619-t002:** Scores of the sensory descriptive attributes and overall acceptability for the samples of the different cultivars studied and for ripening Stages 2 and 3.

	External Appearance	Skin Colour	Flesh Colour	Firmness	Sweetness	Acid	Bitter	Juiciness	Seeds	Fruit Flavour	Overall Acceptability
Cultivars											
DR	6.30	5.42	5.58	6.35	3.76	1.30	*2.93*	5.18	3.33	5.84	6.11
CDB	6.53	6.18	**5.97**	6.80	3.11	1.41	**5.14**	4.71	**4.93**	**6.70**	**6.71**
BT	6.16	*5.04*	*4.02*	5.82	3.97	1.61	4.10	5.00	4.48	*4.05*	*4.37*
SA	5.62	5.85	5.67	5.54	4.09	*1.22*	3.67	5.46	3.81	6.25	6.65
CDN	**7.12 ^†^**	**7.09**	5.23	**7.24**	3.38	**2.32**	3.95	4.39	*2.64*	4.35	5.01
BN	4.92	6.43	5.22	*4.97*	3.46	2.17	3.80	5.49	3.63	4.83	5.08
CE	6.88	6.27	5.09	6.77	3.25	2.11	3.98	4.74	3.67	4.81	5.68
TV	*4.97 ^‡^*	6.24	5.71	5.19	*2.86*	1.61	3.69	*4.39*	3.32	5.91	4.90
BB	5.59	5.15	5.03	5.26	**4.28**	1.84	3.00	**5.56**	4.88	5.21	5.57
Ripening stage											
2	6.24	6.09	5.23	6.19	3.37	1.69	3.61	4.95	3.77	5.10	5.40
3	5.77	5.84	5.33	5.80	3.77	1.77	4.00	5.03	3.93	5.55	5.73
PC ^§^	*0.000*	*0.000*	*0.006*	*0.000*	0.389	*0.032*	*0.008*	0.130	*0.000*	*0.000*	*0.000*
Tukey CI ^¶^	± 1.60	±1.49	±1.64	±1.52	±2.26	±1.10	±1.89	±1.75	±1.77	±1.73	±1.45
PS^#^	*0.037*	0.230	0.684	0.075	0.210	0.651	0.143	0.738	0.526	0.064	0.117

^†^ Numbers in **bold** type indicate the maximum score of the attribute. ^‡^ Underlined numbers in *italics* indicate the minimum score of the attribute. ^§^ Pc: *p*-value for cultivar factor. ^¶^ CI: confidence interval for post-hoc Tukey HSD test. ^#^ Ps: *p*-value for ripening stage factor.

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
