# Peer review of "Evaluation of the Physicochemical and Sensory Characteristics of Different Fig Cultivars for the Fresh Fruit Market"

_foods, 2020, doi:10.3390/foods9050619_

Round 1

Reviewer 1 Report

The article is interesting but it required some corrections. It is quite difficult to read due to a lot of abbreviations (too many). The introduction does not provide a complete state of the art of all the studies conducted so far by several authors on the relationship between chemical-physical parameters and sensory descriptors observed for figs also in relation to volatile compounds, it is rather hasty and many interesting studies are not mentioned.

The major issues are the following:

  • the authors do not indicate the time when the 9 cultivars were harvested: 'De Rey', 'Cuello Dama Blanco', 'Brown Turkey', 'San Antonio', 'Bananas', 'Cuello Dama Negro', 'Colar Elche', 'Tres Voltas L'Any', and 'Blanca Bétera', this is fundamental for their study. I recommend to comment on the differences found in relation to the time of ripening of figs. This is important because it is well known that late varieties (September/October) accumulate more volatile compounds as well as sugars, phenols, carotenoids etc... compared to those at early ripeness, such as those before the first decade of July.

  • In the description of the sensory analysis it is not clear how long, after harvest, the judges and the consumers (age range) evaluated the fruits, moreover, how many sessions during the calendar of more than 3 months of ripening were carried out to evaluate the 9 fig varieties, and finally, the descriptors indicated in this section (external appearance, skin colour, flesh colour, taste, sweetness, sourness, bitterness, juiciness, firmness, and presence of seeds) are less than those showed in table 2.

  • I suggest to insert in the y-axis of figure 1E SST (Brix) or SSC (Brix) and to insert, next to the unit, also the parameters in figure a, b, c, d, f (weight ; width; pH, etc...) .

  • the conclusion paragraph of your work is missing.

Author Response

Response to specific Reviewers' comments:

Authors are thankful to reviewers for the meticulous reviews. We have incorporated the valuable suggestions given by reviewers in the manuscript. On the other hand, the degree of overlap has been significantly reduced in the new version. If they further feel any revision, the suggestions are welcome.

Response to specific Reviewers' comments :

Reviewer #1: The article is interesting but it required some corrections. It is quite difficult to read due to a lot of abbreviations (too many). The introduction does not provide a complete state of the art of all the studies conducted so far by several authors on the relationship between chemical-physical parameters and sensory descriptors observed for figs also in relation to volatile compounds, it is rather hasty and many interesting studies are not mentioned.

Response: Following the reviewer’s suggestion, the introduction has been improved in the corrected manuscript.

The authors do not indicate the time when the 9 cultivars were harvested: 'De Rey', 'Cuello Dama Blanco', 'Brown Turkey', 'San Antonio', 'Bananas', 'Cuello Dama Negro', 'Colar Elche', 'Tres Voltas L'Any', and 'Blanca Bétera', this is fundamental for their study. I recommend to comment on the differences found in relation to the time of ripening of figs. This is important because it is well known that late varieties (September/October) accumulate more volatile compounds as well as sugars, phenols, carotenoids etc... compared to those at early ripeness, such as those before the first decade of July.

Response: We understand the reviewer´s concern. This information has been introduced in the corrected manuscript (Lines 101-106; 246-247; 253-254).

In the description of the sensory analysis it is not clear how long, after harvest, the judges and the consumers (age range) evaluated the fruits, moreover, how many sessions during the calendar of more than 3 months of ripening were carried out to evaluate the 9 fig varieties, and finally, the descriptors indicated in this section (external appearance, skin colour, flesh colour, taste, sweetness, sourness, bitterness, juiciness, firmness, and presence of seeds) are less than those showed in table 2.

Response: Authors fully agree with the reviewer's comment. This aspect has been improved in the corrected manuscript (Lines 194-203).

I suggest to insert in the y-axis of figure 1E SST (Brix) or SSC (Brix) and to insert, next to the unit, also the parameters in figure a, b, c, d, f (weight ; width; pH, etc...) .

Response: Figure has been improved in the corrected manuscript.

The conclusion paragraph of your work is missing.

Response: We apologize for this mistake. The conclusion has been introduced in the corrected manuscript.

Reviewer 2 Report

The article 'Physicochemical and sensorial characterization of different fig cultivars for fresh consumption' presented for review is very interesting. This article has a high research value. The results are well-discussed. However, some issues need to be clarified or supplemented. The comments are included below.

Title

The title is worded correctly and accurately reflects the content.

 Abstract

The abstract is clear and adequate. The length of the abstract is sufficient.

 Introduction

The introduction serves multiple purposes. It introduces well to the subject of the article.

Materials and method

Concerning methodology, the methods used for the experimental part of the study are up to date and relevant. However, some issues need to be clarified or supplemented.

Line 105: Why was the titration to pH 7.8 instead of 8.1? Is this methodology commonly used?

Line 111-113:  Why describe the color parameters a * and b * if they are not in the results discussion.

Results and Discussion

The results are well-discussed. However, some issues need to be clarified or supplemented.

Line  183: It should be (Fig. 1E and F) not (Fig. 1D and E).

Line 186 and line 196: ‘…the more relevant differences…’ - Has statistics been shown in this respect?

Line 224: It should be 68 not 58

Line 233: '…(Trad et al., 2012)…’ - Wrong way of quoting literature. The numbering of articles in the literature will change.

Line 250: It should be 23.05 not 25.03

Conclusion

Not clear what are the conclusions?

Author Response

Response to specific Reviewers' comments:

Authors are thankful to reviewers for the meticulous reviews. We have incorporated the valuable suggestions given by reviewers in the manuscript. On the other hand, the degree of overlap has been significantly reduced in the new version. If they further feel any revision, the suggestions are welcome.

Response to specific Reviewers' comments :

Reviewer #2:

Materials and method

Line 105: Why was the titration to pH 7.8 instead of 8.1? Is this methodology commonly used?

Response: We understand the reviewer´s concern. Titratable acidity was titrated on the basis of citric acid up to the end point of 7.8. This methodology has been previously described in Pereira et al. 2015 (“Agronomic behaviour and quality of six fig cultivars for freshconsumption”-Scientia Horticulturae 185 (2015), 121–128) and Pereira et al. (“Evaluation of agronomic and fruit quality traits of fig tree varieties (Ficus carica L.) grown in Mediterranean conditions”- Spanish Journal of Agricultural Research 15 (2017), e0903, 9 pages).

Line 111-113:  Why describe the color parameters a * and b * if they are not in the results discussion.

Response: Following the reviewer’s suggestion, it has been removed in the corrected manuscript (Line 148).

Results and Discussion

Line  183: It should be (Fig. 1E and F) not (Fig. 1D and E).

Response: We apologize for this mistake. It has been modified in the corrected manuscript (Line 250).

Line 186 and line 196: ‘…the more relevant differences…’ - Has statistics been shown in this respect?

Response: According to confidence interval of Tukey HDS represented in figure 1 F, the differences between ripening stages for the referenced cultivars are statistically significant.

Line 224: It should be 68 not 58

Response: It has been modified in the corrected manuscript (Line 291).

Line 233: '…(Trad et al., 2012)…’ - Wrong way of quoting literature. The numbering of articles in the literature will change.

We apologize for this mistake. It has been modified in the corrected manuscript (Line 304).

Line 250: It should be 23.05 not 25.03

Response: The text is suitably modified now (Line 328).

Conclusion

Not clear what are the conclusions?

Response: We apologize for this mistake. The conclusion has been introduced in the corrected manuscript.

Round 2

Reviewer 1 Report

This manuscript of Pereira  et al. was rationally modified including Materials and Methods and Conclusions after suggested of peer-reviewers.

Amendment "Conclusion" with Conclusions